# Tool Wear Condition Monitoring Method Based on Deep Learning with Force Signals

**DOI:** 10.3390/s23104595

**Published:** 2023-05-09

**Authors:** Yaping Zhang, Xiaozhi Qi, Tao Wang, Yuanhang He

**Affiliations:** 1Shenzhen Institutes of Advanced Technology, Chinese Academy of Sciences, Shenzhen 518055, China; zyping1989@126.com (Y.Z.); xz.qi@siat.ac.cn (X.Q.); 2Institute of Intelligent Manufacturing Technology, Shenzhen Polytechnic, Shenzhen 518055, China; 3State Key Laboratory of Explosion Science and Technology, Beijing Institute of Technology, Beijing 100081, China

**Keywords:** tool wear, deep learning, CNN, force signals

## Abstract

Tool wear condition monitoring is an important component of mechanical processing automation, and accurately identifying the wear status of tools can improve processing quality and production efficiency. This paper studied a new deep learning model, to identify the wear status of tools. The force signal was transformed into a two-dimensional image using continuous wavelet transform (CWT), short-time Fourier transform (STFT), and Gramian angular summation field (GASF) methods. The generated images were then fed into the proposed convolutional neural network (CNN) model for further analysis. The calculation results show that the accuracy of tool wear state recognition proposed in this paper was above 90%, which was higher than the accuracy of AlexNet, ResNet, and other models. The accuracy of the images generated using the CWT method and identified with the CNN model was the highest, which is attributed to the fact that the CWT method can extract local features of an image and is less affected by noise. Comparing the precision and recall values of the model, it was verified that the image obtained by the CWT method had the highest accuracy in identifying tool wear state. These results demonstrate the potential advantages of using a force signal transformed into a two-dimensional image for tool wear state recognition and of applying CNN models in this area. They also indicate the wide application prospects of this method in industrial production.

## 1. Introduction

Automated production processes are an important component of Industry 4.0, and mechanized machining automation is one of the main components of automated production processes. In the machining process, cutting tools are end-effectors that come into direct contact with the workpiece and are prone to wear, which can affect the surface quality of the workpiece [1]. Zhou et al. [2] stated that tool wear and damage are the main factors leading to process failure in machining. Especially for precision high-end machine tools, tool wear problems can lead to low production quality, high labor costs, high maintenance costs, and other issues. Therefore, it is necessary to develop an effective tool monitoring system, to ensure optimal operating conditions of cutting tools, in order to improve machining quality and economy [3].

Tool wear monitoring is a challenging task because many machining processes exhibit nonlinear and time-varying characteristics [4]. Therefore, it can be difficult or impossible to establish an accurate theoretical model for precise monitoring, and it is also difficult to directly measure tool wear during the cutting process. Generally speaking, tool wear monitoring can be divided into two categories: direct monitoring of machining processes, and indirect monitoring of cutting tools. Tool condition monitoring through direct methods usually involves using charge-coupled device (CCD) cameras to capture the actual geometric changes caused by tool wear, allowing for non-contact observation of the cutting tool itself. Directly evaluating cutting tools using machine vision is a challenging task because in the machining process, whereby the cutting zone cannot be accessed due to continuous contact between the tool and workpiece, as well as the presence of cooling fluids and chip obstruction [5]. In addition, some direct tool condition monitoring methods require the separation of the tool from the holder, which may result in tool misalignment during the next operation [6]. As a result, most direct measurement techniques are limited to laboratory technologies, with very few applicable to industrial applications.

With the advancement of sensor technology, indirect monitoring methods are becoming increasingly important. These methods use one or more sensors to obtain physical signals such as cutting force [7,8], acoustic emission (AE) [9], motor current [10,11], vibration [12], and sound [13] signals, to extract features related to tool condition [14]. Afterwards, certain artificial intelligence (AI) methods are used to detect tool wear, such as multiple regression [15], simple neural networks [16], support vector machines [17], random forests [18], hidden Markov models [19], etc. However, these traditional methods often face the following problems: shallow machine learning models, such as neural networks and support vector machines, can achieve good results, but they often require a lot of prior knowledge and repeated validation in the data processing stage, which is inefficient and time-consuming for modeling.

By combining industrial big data with deep learning (DL), repeatedly training deep learning models on production data can improve their generalization ability and accuracy. By analyzing the potential connections between collected data and machines, this approach can guide industrial production planning, improve efficiency, and reduce production costs. Typical deep learning models include an input layer, several hidden layers, and an output layer, with multiple neurons in each layer that are fully interconnected between adjacent layers. This end-to-end black box approach is particularly useful for tool condition monitoring (TCM) in milling processes, as the relationship between sensor signals and tool wear during milling is nonlinear and difficult to express using analytical formulas. In addition, time series signals can be directly input into DL models, to learn deep features without the need for feature extraction [20,21,22]. Li et al. [23] transformed sound signals into two-dimensional images via short-time Fourier transform and developed a novel integrated convolutional neural network (CNN) model for intelligent and accurate recognition of tool wear. The proposed method improved the tool wear detection performance by at least 1.2%. Zhou et al. [24] fused sound signals in three different channels and proposed a new improved multi-scale edge-aware graph neural network to improve the identification accuracy of TCM, based on deep learning under small-sample conditions. Tran et al. [25] used continuous wavelet transform to convert cutting force signals into two-dimensional images, and combined these with CNN models to effectively detect chatter occurrence. Li et al. [26] transformed vibration signals into two-dimensional images via short-time Fourier transform, used convolutional neural networks to extract multi-scale features, and accurately predicted the remaining life of the tool.

However, to improve the training accuracy of deep learning models, a large amount of input data are often required. Unfortunately, in many industrial scenarios such as TCM, collecting large samples requires high levels of funding and extensive experimental time, which can be challenging for many manufacturing companies. In addition, it is not sufficient to build an effective and reliable tool wear monitoring system through conventional signal processing without computer vision techniques [27]. Existing research [28,29] results suggest that it is necessary to incorporate computer vision techniques into the audio analysis framework for tool wear, in order to generate reliable tool wear detection results.

In order to solve the above problems in tool wear monitoring, this paper first converts the force signal into a two-dimensional image through continuous wavelet transform (CWT), short-time Fourier transform (STFT), Gramian angular summation field (GASF), etc. Furthermore, this paper proposes a CNN model with a multi-scale feature pyramid that takes into account both high-level and low-level features of the images obtained after signal transformation. The high-level features and low-level features are fused together, and the high semantic information is passed down to the low-level features, significantly improving the accuracy of tool wear monitoring.

## 2. Research Methods

### 2.1. Continuous Wavelet Transform (CWT)

CWT is a time–frequency analysis method that can convert one-dimensional signals in the time domain into two-dimensional representations in the time-frequency domain with high resolution. CWT is an extension of wavelet transform and decomposes time-domain signals at different scales, while calculating the time-varying frequency spectrum at each scale. CWT is effective for analyzing transient features in non-stationary signals and thus widely used in fault diagnosis, remaining-life prediction, and other fields of mechanical condition monitoring. The CWT of a signal, *x*(*t*), is defined in Equation (1) [30].
(1)$$W_{\psi x\left(s,\;u\right)}=\frac{1}{\sqrt{s}}\;\int_{-\infty }^{+\infty }x\left(t\right)\psi \left(\frac{t-u}{s}\right)\mathrm{dt}$$

Here, *u* represents the translation parameter and *s* is the scaling parameter of the wavelet function *ψ*(*t*). CWT provides a variable resolution in the time–frequency plane. For high-frequency components, CWT can achieve a high time resolution and low frequency resolution; while for low-frequency components, it can obtain a high frequency resolution and low time resolution. This is because as s increases, the frequency of the wavelet function decreases and the time resolution decreases. Conversely, as u increases, the frequency of the wavelet function increases and the frequency resolution decreases. Therefore, by appropriately selecting the wavelet function and its parameters, transient features of non-stationary signals can be effectively analyzed in the time–frequency domain.

There are several types of wavelet family, and each type of wavelet function has a specific smoothness, shape, and compactness, giving them different applications. The Morlet mother wavelet is a complex exponential wavelet with a Gaussian envelope that is widely used in various applications. Research has shown that the Morlet wavelet function can provide good time and frequency resolution [31]. Therefore, in this study, the Morlet wavelet function was used to analyze cutting force signals with non-stationary and nonlinear behaviors.

### 2.2. Short-Time Fourier Transform (STFT)

STFT is a time–frequency analysis method that is similar to CWT. It can also transform time-domain signals into images with high resolutions in both the time and frequency domains. Based on the Fourier transform, STFT segments the signal into multiple windows and performs Fourier transform on each window, to obtain the frequency spectrum within that window. Unlike CWT, the windows in STFT are independent of each other, which means it cannot handle transient features in non-stationary signals. However, due to its simple computation and fast speed, STFT is widely used in many applications. STFT is a commonly used method for discovering information about non-stationary signals in both the time and frequency domains [32]; by applying the STFT algorithm to the signal *x*(*t*) to transform it.
(2)$$I\left(\tau ,\;\omega \right)=\int_{-\infty }^{+\infty }x\left(t\right)w_{1}\left(t-\tau \right)e^{-j\omega t}\mathrm{dt}$$

Here, $w_{1}\left(t-\tau \right)$ is the window function centered at time *τ*. For each different time, STFT generates a different spectrum, and the sum of these spectra forms the spectrogram.

### 2.3. Gramian Angular Summation Field (GASF)

As a preprocessing step, our method employs the GAF imaging technique [33], particularly the angle-based summation method known as Gramian angular summation field (GASF). GASF is a novel time–frequency analysis method based on a Gramian matrix, which has emerged in recent years. Unlike CWT and STFT, a GASF does not require time–frequency decomposition of the signal. Instead, it converts the time series into an angle sequence, computes the cosine values to obtain the Gramian matrix, and then adds its elements to obtain the GASF. GASF can effectively capture the periodicity and contour features of the signal, thus it has been widely used in signal classification, pattern recognition, and other fields. This encoding method consists of two steps. First, the time series is represented in a polar coordinate system instead of the typical Cartesian coordinate system. Then, as time increases, the corresponding values on the polar coordinate system rotates between different angular points on the circle. This representation preserves the temporal relationships and can be easily used to identify time correlations across different time intervals. This time correlation is expressed as
(3)$$G=\left[\begin{array}{ccc} \cos\left(\phi_{1}+\phi_{1}\right) & \cdots & \cos\left(\phi_{1}+\phi_{n}\right)\\ \vdots & \ddots & \vdots \\ \cos\left(\phi_{n}+\phi_{1}\right) & \cdots & \cos\left(\phi_{n}+\phi_{n}\right) \end{array}\right]$$

ϕ is the polar coordinate representation of the time series X. The GASF image provides a way to preserve temporal dependencies. As the position in the image moves from the upper left to lower right, time increases. When the length of the time series is *n*, the resulting GASF image has dimensions of *n* × *n*. To reduce the size of the image, piecewise aggregate approximation (PAA) is applied to smooth the time series, while preserving the trends.

### 2.4. The CNN Model with Multiscale Feature Pyramid

The multiscale feature pyramid scheme is a technique used in computer vision for object detection and recognition tasks. The basic idea behind this scheme is to create a set of feature maps at multiple scales that can be used by object detection algorithms to detect objects of different sizes and shapes in an image.

The mechanism of the multiscale feature pyramid scheme involves the use of a convolutional neural network (CNN) that extracts features from an input image at multiple scales. These features are then processed using a set of additional convolutional layers, to create a set of feature maps at different scales. The feature maps at each scale are created by applying a series of convolutions, followed by pooling operations that downsample the spatial dimensions of the feature map.

The benefit of the multiscale feature pyramid scheme is that it allows object detection algorithms to detect objects of different sizes and shapes in an image. By using feature maps at multiple scales, the algorithm can detect small objects that appear in large images, as well as larger objects that appear close to the camera. Another benefit of the multiscale feature pyramid scheme is that it allows for efficient computation. The feature maps at different scales can be computed in parallel, which reduces the computational cost and speeds up the detection process. Additionally, the use of pooling operations to downsample the feature maps reduces the number of parameters required by the algorithm, which can improve its efficiency and reduce the risk of overfitting.

Considering the continuity of the machining process, the wear pattern of the cutting tool after machining will determine the accuracy of the next machining. Therefore, this paper proposes a CNN model with a multiscale feature pyramid that integrates the low-dimensional and high-dimensional features of the image, and passes the high-dimensional semantic features to the lower layer features. This fusion method of high- and low-dimensional features allows the model to obtain rich visual and semantic characteristics in low-dimensional feature extraction of an image. A specific overview is shown in Figure 1.

The two-dimensional image transformed by CWT is used as the input to the feature pyramid, and it undergoes three convolution operations. Convolution 1 uses a 1 × 1 kernel with a rectified linear unit (ReLU) activation function; Convolution 2 uses a 1 × 1 kernel with ReLU activation function, followed by a 3 × 3 convolution operation with a ReLU activation function; and Convolution 3 uses a 1 × 1 kernel with ReLU activation function, followed by a 5 × 5 convolution operation with a ReLU activation function. Then, the results of Convolution 1 and Convolution 2 are fused, and the fused result is combined with Convolution 3 to obtain the final input. The resulting output is passed through two convolutional layers and one linear layer to obtain the final classification result.

## 3. Force Signal to Image

To obtain more accurate tool wear classification information, the Prognostics and Health Management (PHM) 2010 milling TCM dataset [34] was used to verify the practicality of the proposed CNN model. This dataset applies a three-component dynamometer to measure cutting force signals in three directions (X, Y, Z). Three piezoelectric accelerometers and one AE sensor were installed on the workpiece to measure vibration and stress waves in these three directions. Optical microscopes were used to measure the corresponding flank wear of each flank of the milling tool.

The PHM 2010 milling TCM dataset includes six groups of data, namely C1, C2, C3, C4, C5, and C6. Among them, C1, C4, and C6 are labeled datasets, i.e., the collected signals have corresponding tool wear values after each machining operation. In this paper, the C1 and C4 datasets were taken as the training set, and the C6 dataset was used as the validation set. Taking the C4 dataset as an example, the curve of its wear value with the number of cutting times n is shown in Figure 2. The curve can be divided into three segments: the initial wear stage (cutting 0–26 times), the stable wear stage (cutting 27–208 times), and the sharp wear stage (cutting 209–315 times). Based on this, the Z-direction force signals in the C4 dataset were divided into three categories according to the wear stage.

### 3.1. Force Signal Description

Figure 3 shows the variation of the Z-axis force signal over time at different stages of wear. It can be observed that the amplitude and period of force changes are different at different tool wear stages. In the initial wear stage, the vibrational fluctuation of the force is not too large, and the maximum value of the force is around 10 N. During the machining process, the contact between the tool and the workpiece can cause an increase in vibration amplitude of the force signal. At the end of the machining process, the force rises to about 90 N. Typically, there may be some noise during signal acquisition; however, the force signal is less affected by noise compared to other signals. This is also one of the reasons why the force signal was chosen as the research object in this paper. When converting a signal into an image, it is usually not necessary to use the entire signal as input. Only a small segment of stable machining (about 0.1 s) is considered, as shown by the black dashed box in Figure 3. Amplifying the signal within this box yielded Figure 4. It can be seen that the vibration amplitude of the force signal is relatively regular and can fully reflect the changing pattern of the entire signal segment. There is no significant jumping up and down, indicating that the force signal is not affected by noise and thus does not require any noise reduction processing.

### 3.2. Force Signal to Image

Figure 5 shows a two-dimensional image of the force signal in Figure 4, which was converted using the continuous wavelet transform (CWT) method. It can be seen that the image characteristics vary with different tool wear stages. In total, 630 images were generated by applying the CWT method to the force signals in the C1 and C4 datasets, and then these images were divided into three categories according to their different tool wear stages, as the training set. Similarly, the force signals in the C6 dataset were transformed into images, generating a total of 315 images, which were also divided into three categories according to their different tool wear stages, as the validation set.

Before using the STFT method, it is necessary to perform amplitude–frequency analysis of the input signal, to determine the energy interval of the signal. Figure 6 shows an amplitude–frequency diagram of the force signal after the fast Fourier transform (FFT). Throughout the machining process, the energy is concentrated in the low-frequency region. However, during the initial wear stage, the energy is high and concentrated near 500 Hz; during stable cutting, most of the energy is distributed in the low-frequency region, around the tooth passing frequency and its harmonics. In the severe wear stage, the energy gradually shifts to the high-frequency region. Due to the presence of cutting chips, cutting fluid, and other factors, some noise may be added to the signal acquisition, which can be observed in Figure 6c with a large amount of vibration energy around 1200 Hz. Therefore, the three different cutting states can be detected by classifying the proportional image graphs.

After obtaining the amplitude–frequency plot of the signal, the STFT method was used to convert the force signal into a two-dimensional image. Figure 7 is the image generated after STFT conversion. From Figure 7, it can be seen that all the features of the image generated by STFT are in the lower part of the image, which is consistent with the conclusion obtained from the FFT magnitude in Figure 6; that is, the energy is concentrated in the low-frequency region. With the wear of the cutting tool, the image features gradually become clearer, and finally, the image only has a few clear horizontal line features concentrated in the low-frequency band. The division of the training set and validation set was consistent with the above.

To create a GASF representation of a time series, the time series values are first normalized to fall between −1 and 1. Then, the cosine and sine of each value in the time series are calculated, and these values are used to create a Gram matrix. The Gram matrix is further transformed to create a GASF image, which represents the distribution of angles between pairs of values in the original time series.

The resulting GASF image provides a visual representation of the underlying patterns and structures within the time series, which may not be apparent from the raw data alone. This can be useful in identifying similarities and differences between different time series, and for detecting patterns and anomalies within a single time series.

The GASF image is related to the number of pixels that need to be set. In order to maximize the retention of image features and avoid unnecessary computation, the selected pixel count in this article was 128 × 128. As shown in Figure 8, the GASF-generated image had rich image features, and there were significant differences in the image information between the different wear stages. The division of the training set and validation set was consistent with the above.

To provide a more intuitive understanding of the three methods for generating images, Figure 9 shows the images generated by the three methods for the same signal. Comparing Figure 9b, Figure 9c and Figure 9d it is clear that the images generated by the three methods are very different. The image generated by the CWT method provides more comprehensive and detailed information about the signal, helping viewers to better understand and analyze the signal. The image generated by the STFT method has fewer features and is concentrated in the low-frequency region. Images generated by the GASF method usually exhibit symmetry and periodicity, but due to their complexity, further processing and analysis may be needed to draw meaningful conclusions.

## 4. Analysis Procedure

### 4.1. Results and Analysis of the Datasets

The architecture of a convolutional neural network typically consists of an input layer, several hidden layers, and an output layer. The image obtained in the previous paragraph was used as the input layer of the convolutional neural network, with a pixel size of 128 × 128 × 3; where 128 is the number of pixels in a single image, and 3 indicates that the image is in color. The hidden layer section consists of three convolutional layers (Conv1, Conv2, and Conv3), as well as their fusion layer. Following that, there are two more convolutional layers and a linear layer. The output layer has three classes. To prevent overfitting, the dropout function was added to the convolutional layers. The learning rate of the neural network was also set to Learning rate (*lr*) = 0.001, and the optimization algorithm used was stochastic gradient descent with momentum (SGDM), which is typically used for image classification. The *momentum value* was set to 0.9 and the *learning rate decay* was set to 0.005. As this was a multi-class classification problem, the loss function used was the CrossEntrophy loss function, and the probability of each class was output using a Softmax function. The specific parameters can be seen in Table 1.

Using the CNN model proposed in this article for training and validation, the accuracy of three generation methods is shown in Figure 10. All three methods had an accuracy greater than 90% during training, with the accuracy when using the CWT method reaching 95%. Although the accuracy of using the GASF method reached nearly 99% during training, its validation accuracy was not as high, only reaching around 80%. The training accuracy of using the STFT method to generate images was 92%. The training accuracy of the three methods was relatively close, but slightly different. This could be determined from the features of the generated images themselves. Time–frequency methods such as CWT can obtain more effective two-dimensional information than GASF, without additional computational cost. As shown in Figure 5, the images generated using the CWT method could be clearly divided into three categories. The GASF method generated images by generating a random matrix, which produces images with rich features, but may have some noise. Considering that the accuracy started to oscillate after 150 epochs, this indicates that the model was overfitting, and that increasing the number of epochs may not further improve the accuracy. In comparison, the STFT method is a time-domain-based image feature extraction method that can obtain relatively stable features, without the need to generate random matrices. However, the image features obtained by the STFT method were limited, and it may be more difficult to distinguish between different categories. As a result, the training accuracy and validation accuracy were largely overlapping, with little difference in accuracy between them.

Figure 11 shows the change curves of the training and validation loss values of the model. The change curve of the loss value corresponds to the accuracy curve in Figure 11. The training and validation loss values of the image dataset transformed by the CWT method were highly consistent, and they started to oscillate and reach equilibrium after 100 epochs. The training loss of the image dataset obtained by the GASF method was close to 0, but its validation error was relatively large, and the difference between the two was the largest among the three methods.

By comparing the accuracy of the model with the existing literature, it can be shown that the CNN model proposed in this paper had a high accuracy when considering high- and low-level features, fusing high- and low-level feature maps, and outputting feature maps with rich semantics. Table 2 lists the prediction accuracies of common deep learning methods in TCM. It can be seen that the CNN model proposed in this paper had the highest accuracy, once again proving that determining the wear state of cutting tools cannot rely on considering single image information only, but also needs to consider the high- and low-dimensional features.

### 4.2. Precision and Recall

Precision and recall are two metrics that can comprehensively evaluate the performance of a model in different categories, especially in the case of imbalanced datasets. In such cases, using only accuracy as an evaluation metric may lead to the model focusing too much on categories with a larger number of samples, while ignoring categories with fewer but more important samples. Using precision and recall can better evaluate the model’s performance in each category, and thus provide a more comprehensive evaluation of the model’s performance.

Precision refers to how many of the samples predicted as true are actually true. The formula for calculation is:Precision = TP/(TP + FP)(4)

TP stands for true positive, indicating that the sample is actually positive and that the model also predicts it to be positive. This is the correct prediction part.

FP stands for false positive, which means that the sample is actually negative, but the model predicts it as positive. This is the incorrect prediction part.

Recall refers to how many true positive samples were correctly identified. The formula for calculating recall is:Recall = TP/(TP + FN)(5)

FN stands for false negatives, which means that the sample is actually positive, but the model predicts it as negative. This is the part of the prediction that is incorrect.

Figure 12 shows the precision of the entire model, and it can be seen that the image obtained using the CWT method had the highest precision, which verifies what was previously stated. This may have been due to the fact that the CWT method can perform multi-scale decomposition on images, making image features more comprehensive and allowing the capture of information at different scales; the CWT method can extract the local features of images, such as edges and textures, and classify and recognize these features; and the CWT method has good noise resistance. It should be noted that precision alone cannot indicate the performance of a model, because this only considers the ratio of correctly predicted positive samples to all predicted positive samples and does not take into account negative samples. Therefore, other indicators such as recall rate need to be comprehensively considered to evaluate the overall performance of a model.

Figure 13 shows the recall of the entire model. It can be seen that the image obtained after STFT had the highest recall rate of 93.33%, indicating that this method could identify most true positives when predicting tool wear state, but with a relatively low accuracy. This may have been because the image features generated by the STFT method were relatively few, making it difficult for the classifier to distinguish them from other images.

Table 3 lists the precision and recall values of common deep learning methods in TCM. Through a series of comparative experiments, precision and recall values were obtained for the CNN model, the ResNet model, and the AlexNet model. By comparing these values with our model (using images generated by the CWT method as an example), it can be seen that our model’s precision value was significantly higher than that of other models; by 6% compared to the CNN model, 33% compared to the ResNet model, and nearly 30% compared to the AlexNet model. Similarly, our model also achieved comparatively good results for the recall metric.

The proposed CNN model with a multiscale feature pyramid is capable of accurately classifying tool wear states based on force signal data, which is a novel application of deep learning in the field of manufacturing. By converting force signals into two-dimensional images, the proposed approach provides a new perspective for tool wear monitoring and enables the use of computer vision techniques for accurate classification. Compared to traditional methods that rely on large amounts of sample data, the proposed method requires relatively small amounts of training data, making it more cost-effective and practical for industrial applications. The multiscale feature pyramid architecture used in the CNN model is a novel design, which effectively captures features at different scales, improving the accuracy of the classification results.

Overall, the technical contributions and novelty of the current study demonstrate its significance in advancing the field of tool wear monitoring and providing an efficient and effective solution for industrial applications.

## 5. Conclusions

This study proposes a novel method for recognizing the wear condition of cutting tools based on converting force signals into two-dimensional images. A deep convolutional neural network (CNN) model was developed that considers both high- and low-dimensional features of images, to identify the degree of tool wear intelligently and accurately. First, the collected force signals are transformed into corresponding two-dimensional images using methods such as continuous wavelet transform (CWT), short-time Fourier transform (STFT), and Gabor feature analysis (GASF). Subsequently, these images are fed into the proposed CNN model. Through calculations on the PHM 2010 TCM dataset, the proposed method achieved excellent recognition results. The results show that (1) using the CNN method proposed in this paper, the accuracy of tool wear classification is significantly improved and can reach over 90%. Compared to traditional methods that rely on large amounts of sample data, our proposed method requires relatively small amounts of training data, making it more cost-effective and practical for industrial applications. (2) Using the image dataset generated by the CWT method achieved the highest accuracy in identifying tool wear categories. (3) The accuracy of the CNN model proposed in this paper was 10% higher than existing methods such as AlexNet, ResNet, and MEGNN. (4) Comparing the precision and recall of the models, it was confirmed that the CWT method yielded the highest accuracy in predicting the tool wear state based on the obtained images. Using a multiscale feature pyramid architecture for tool wear state classification is innovative and can improve classification accuracy. Overall, this method has significant practical applications and can provide cost-effective solutions to improving manufacturing processes. In the future, our plans include further improvements in signal-based tool wear monitoring methods, to mitigate the effects of data imbalances. We will also conduct research on predicting tool wear for timely maintenance and explore online monitoring of real machine tools.

## Figures and Tables

**Figure 1 sensors-23-04595-f001:**
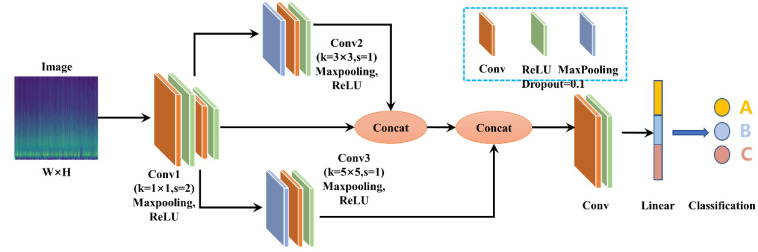
Overview of the proposed tool condition monitoring method based on CNN. A, B, and C represent the initial wear, stable wear, and sharp wear stages, respectively.

**Figure 2 sensors-23-04595-f002:**
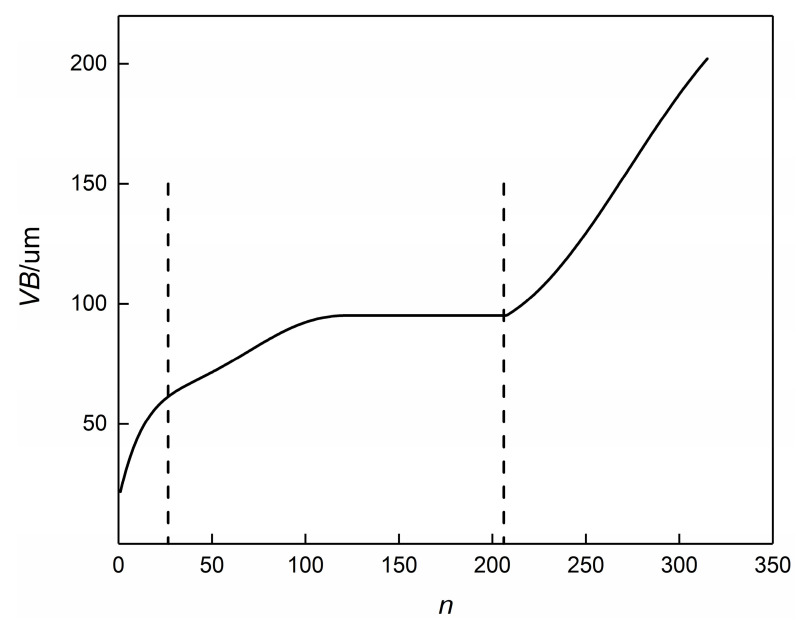
Curve of the change in tool wear *VB* value with the number of cutting times. The dashed line represents the division of the machining process into three different stages, namely, the initial wear stage, the stable wear stage, and the sharp wear stage.

**Figure 3 sensors-23-04595-f003:**
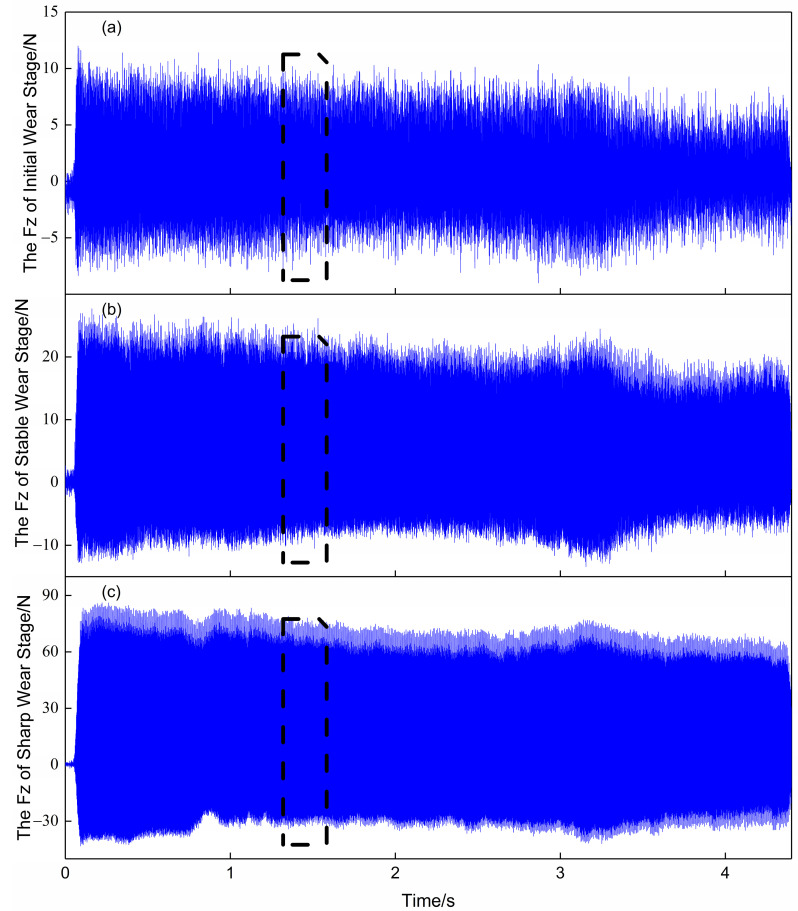
Curves showing the variation of force signals with time at different wear stages. (**a**), (**b**), and (**c**) represent the Z-direction force signals in the initial wear, stable wear, and sharp wear stages, respectively. The dashed boxes represent only a small segment of stable machining (about 0.1 s).

**Figure 4 sensors-23-04595-f004:**
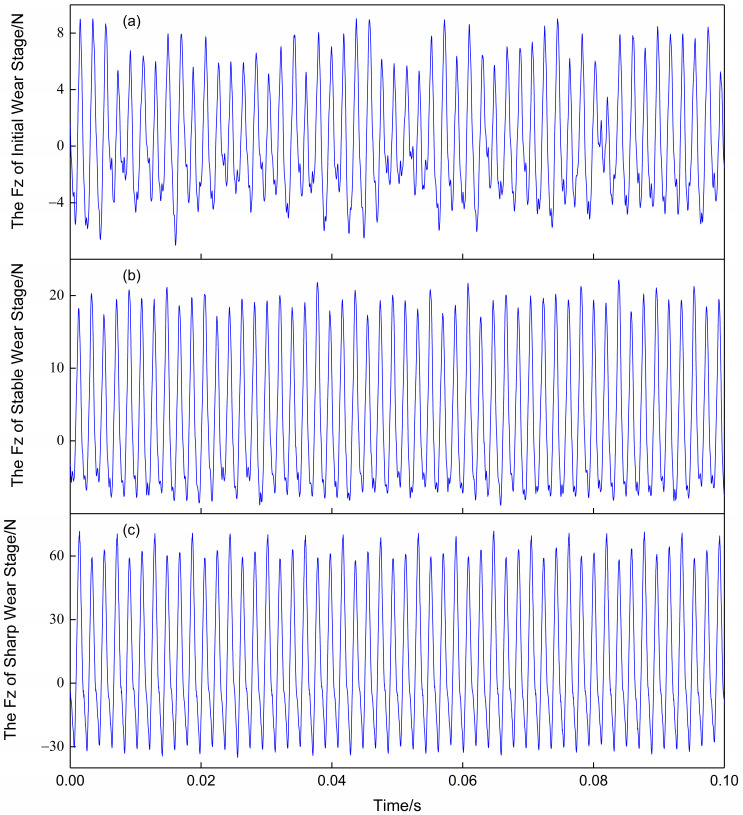
Temporal evolution curve of force signal within 0.1 s in different wear stages. (**a**), (**b**), and (**c**) represent the Z-direction force signals in the initial wear, stable wear, and sharp wear stages, respectively.

**Figure 5 sensors-23-04595-f005:**
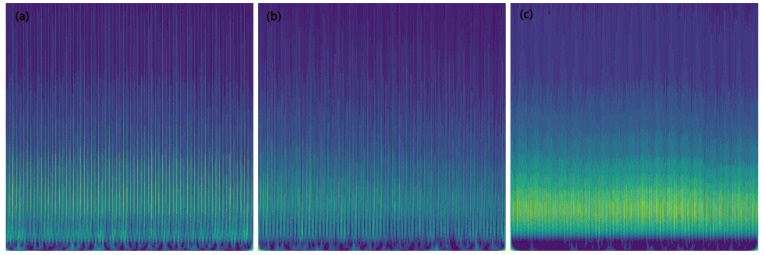
Using the CWT method to convert force signals into two-dimensional images at different wear stages. (**a**), (**b**), and (**c**) represent the image at the initial wear, stable wear, and sharp wear stages, respectively.

**Figure 6 sensors-23-04595-f006:**
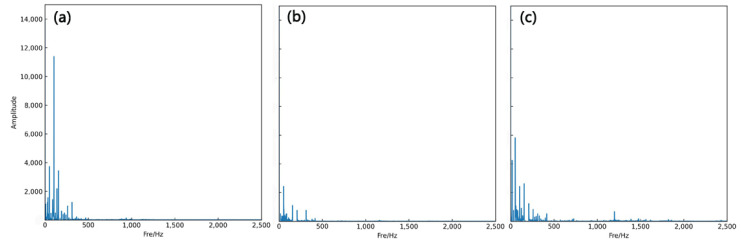
FFT magnitude. (**a**), (**b**), and (**c**) represent the image during the initial wear, stable wear, and sharp wear stages, respectively.

**Figure 7 sensors-23-04595-f007:**
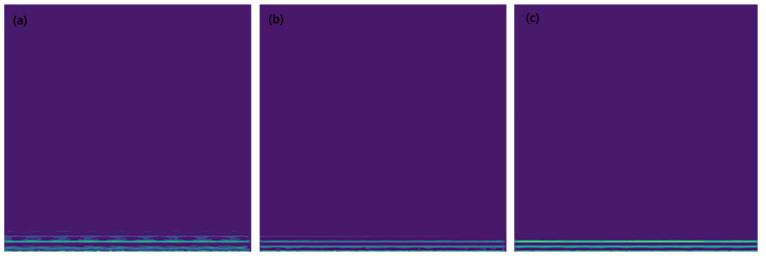
Using the STFT method to convert force signals into two-dimensional images during different wear stages. (**a**), (**b**), and (**c**) represent the image in the initial wear, stable wear, and sharp wear stages, respectively.

**Figure 8 sensors-23-04595-f008:**

Using the GASF method to convert force signals into two-dimensional images during different wear stages. (**a**), (**b**), and (**c**) represent the image in the initial wear, stable wear, and sharp wear stages, respectively.

**Figure 9 sensors-23-04595-f009:**
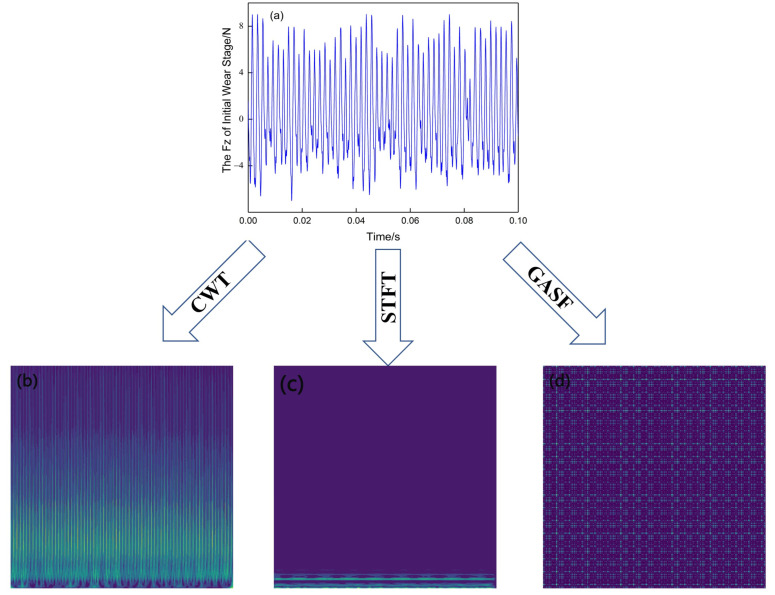
Generating a 2D image from the same single signal. (**a**), (**b**), (**c**) and (**d**) represent the signal, image generated by the CWT method, image generated by the STFT method, and image generated by the GASF method, respectively.

**Figure 10 sensors-23-04595-f010:**
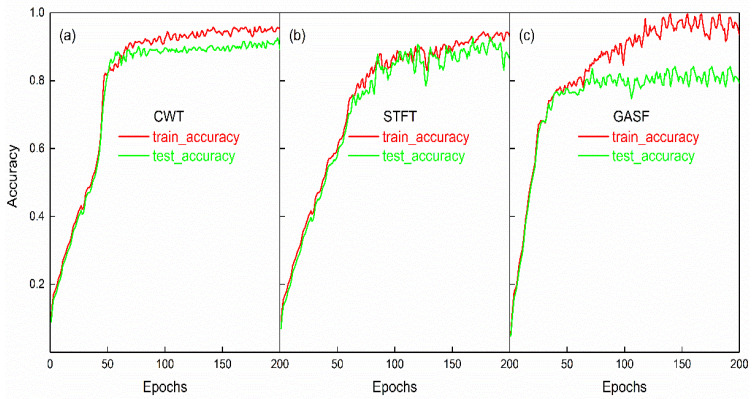
The accuracy of the CNN model, (**a**), (**b**), and (**c**) represent the image datasets generated by CWT, STFT, and GASF methods, respectively. The red and green lines represent the training accuracy and validation accuracy, respectively.

**Figure 11 sensors-23-04595-f011:**
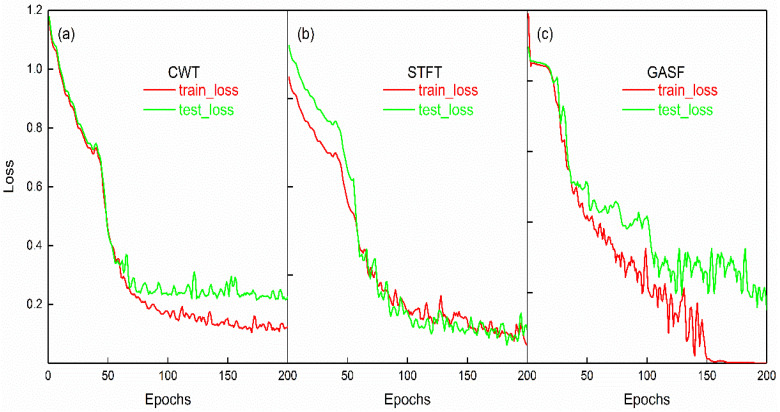
The loss item of the CNN model, (**a**), (**b**), and (**c**) represent the image datasets generated by CWT, STFT, and GASF methods, respectively. The red and green lines represent the training loss and validation loss, respectively.

**Figure 12 sensors-23-04595-f012:**
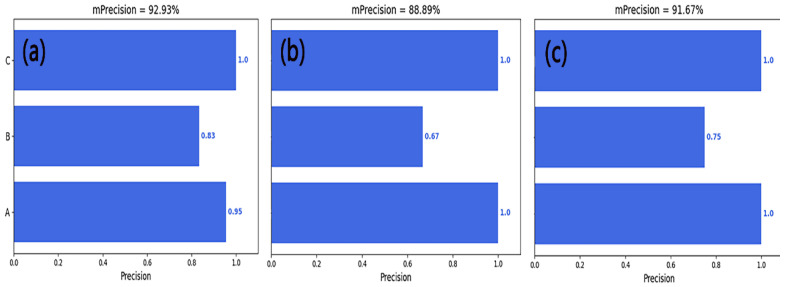
The precision of the CNN model, (**a**), (**b**), and (**c**) represent the image datasets generated by CWT, STFT, and GASF methods, respectively. A, B, and C correspond to three types of tool wear: A represents the initial wear stage, B represents the stable wear stage, and C represents the sharp wear stage.

**Figure 13 sensors-23-04595-f013:**
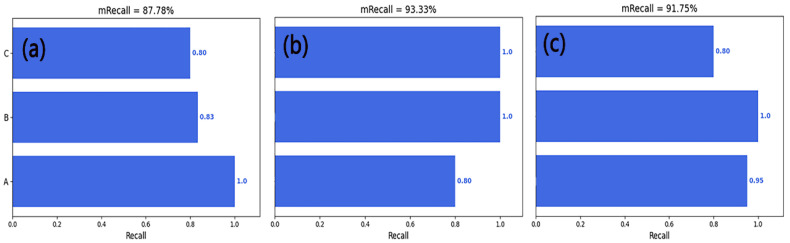
The recall of the CNN model, (**a**), (**b**), and (**c**) represent the image datasets generated by CWT, STFT, and GASF methods, respectively. A, B, and C correspond to three types of tool wear: A represents the initial wear stage, B represents the stable wear stage, and C represents the sharp wear stage.

**Table 1 sensors-23-04595-t001:** Parameters of the CNN model.

Parameter	Image Size	Learning Rate	Dropout	Batch Size	Optimizer	Loss Function
Value	128 × 128 × 3	0.001	0.1	4	SGDM	CrossEntrophy

**Table 2 sensors-23-04595-t002:** Classification accuracy with different deep learning models for TCM.

	CNN	ResNet	AlexNet	MEGNN	Ours
Accuracy	89%	62.9%	70.1%	80.8%	>90%
Source	Ref. [33]	Ref. [24]	Ref. [24]	Ref. [24]	This paper

**Table 3 sensors-23-04595-t003:** Precision and recall values with different deep learning models for TCM.

	CNN	ResNet	AlexNet	Ours (CWT)
Precision	86.7%	59.7%	64.7%	92.93%
Recall	83.2%	55.9%	61.3%	87.78%
Source	Ref. [33]	Ref. [24]	Ref. [24]	This paper

## Data Availability

Not applicable.

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
