# Peer review of "Tool Wear Condition Monitoring Method Based on Deep Learning with Force Signals"

_sensors, 2023, doi:10.3390/s23104595_

Round 1

Reviewer 1 Report

1. Please define the full name of terminology before using the abbreviation. This issue observed throughout the manuscript. Please rectify.

2. References should be cited for sentence of "Existing research results suggest that it is necessary to incorporate computer vision techniques into the audio analysis framework for tool wear to generate reliable tool wear detection results.". Also, how does the proposed method is relevant with this statement? Please clarify.

3. Technical contributions and novelty of current study should be highlighted to justify the significance of current study.

4. Please ensure all parameters and variables are presented in italic format.

5. Section 2.4 - Please elaborate the mechanisms of multiscale feature pyramid scheme employed and explain the benefits of this scheme. 

6. Figure 1 - What classes to be classified by output nodes assigned with orange, blue and red colors? Please assign accordingly in the figure to make it more self-explained. 

7. Table 2 - What are the input image used for "Our Model"? Is it generated based on input based on CWT, STFT, or GASF?

8. Authors should conclude the proposed deep learning framework can deliver the best results based on which transformation technique. The proposed deep learning should also be compared with other deep learning methods in terms of precision and recall values. The quantitative results of these metrics should be presented. 

9. TCM datasets of C1 and C2 are used as training set, while C6 is used as validation set. How about C3, C4 and C5? Are these three datasets play any roles in the proposed study?

Quality of English Language in most sections are acceptable. Some improvements are needed for Section 4.2 because the quality looks different with the rest of sections. Please define the full name of terminology before using the abbreviation. This issue observed throughout the manuscript. 

Reviewer 2 Report

please, see the file attached.

Reviewer 3 Report

This paper proposes a novel method for recognizing the wear condition of cutting tools based on converting force signals into two-dimensional images. The cutting force signal is transformed into a two-dimensional image using methods, i.e., CWT, STFT, and GASF. Afterwards, the generated images are fed into the proposed CNN model to achieve tool wear condition monitoring. Overall, the subject of the paper is interesting, but its weakness lies at the explanation about the novelty. My comments are as follows:

1. Some corrections should be carried out on English writing and grammar. Section 2.2 on page 3: “By applying the STFT to transform the signal x(t)” has a grammar error. And in Figure 1, “the proposed tool conditional monitoring method” should be “the proposed tool condition monitoring method”. The authors can check and further polish the manuscript.

2. The literature review of this paper should be enriched to enhance the introduction part of the manuscript, such as, 10.1177/14759217221088492; 10.3390/s20216113; 10.3390/s22218343; 10.3390/s21010108; 10.1088/1361-665X/acb2a0.

3. The cutting force signals in three different directions are collected by PHM experiment. The author should explain why the force signals in Z direction are chosen for analysis.

4. The author should explain the novelty of this proposed method in comparison with the previous studies and highlight the contributions of the work in conclusion section.

The authors can check and further polish the manuscript.

Round 2

Reviewer 1 Report

All my previous comments have been addressed by the authors accordingly. However, I would like to advise the authors to resubmit the revised manuscript based on the MDPI journal template. 

No issue with the English language quality now. Authors have made some improvements.

Reviewer 2 Report

My comments were taken into account. Accepted.

Reviewer 3 Report

They have properly addressed all my comments.